

# Unveiling the immune and vitamin profiles of blood: the potential biomarkers for alopecia areata

Jincheng Ke[1,2,3], Fangfang Chen[2,4], Yu-Pei Chen[2,4,5], Mingli Zhang[1] and Li Ma[1]

[1] Department of Dermatology, The Second Affiliated Hospital of Xiamen Medical College, Xiamen, Fujian, China
[2] Engineering Research Center of Natural Cosmeceuticals College of Fujian Province, Xiamen Medical College, Xiamen, Fujian, China
[3] The First Hospital of China Medical University, Shenyang, Liaoning, China
[4] School of Public Health and Medical Technology, Xiamen Medical College, Xiamen, Fujian, China
[5] School of Public Health, Fujian Medical University, Fuzhou, Fujian, China

Corresponding author
Yu-Pei Chen,
201600080006@xmmc.edu.cn

## ABSTRACT

**Background:** Alopecia areata is a hair follicle disorder characterized by the development of multiple circular bald patches on the scalp, often accompanied by elevated cytokine production and immune cell infiltration around hair follicles. Our aspiration is to explore whether blood analysis can reveal additional factors that contribute to the disparities between individuals with alopecia areata and those who are healthy. Such research could potentially establish a robust foundation for the advancement of future therapeutic strategies.

**Methods:** In Fujian, China, we have collected blood samples from a cohort of 28 alopecia areata patients and a control group of 28 healthy individuals for comparative analysis. A detailed assessment of cytokines, eosinophil counts, vitamin levels, and immunoglobulin profiles within these samples was conducted. Subsequently, statistical analysis was applied to elucidate the differences between the two groups.

**Results:** While the blood analysis revealed higher average levels of IL-4, IL-10, and TNF-α in alopecia areata patients compared to healthy individuals, these differences were not statistically significant. Similarly, vitamin levels showed no significant variation between the patient and healthy groups. However, the Wilcoxon rank sum test identified a significant increase in IFN-γ and a significant decrease in immunoglobulin IgG4 levels among alopecia areata patients, pointing to a possible role in the disease's pathogenesis. Receiver operating characteristic (ROC) curve analysis had demonstrated that the area under the ROC curve (AUC) for IFN-γ and IgG4 was 0.656 and 0.704, respectively, suggesting that IFN-γ and IgG4 had a certain discrimination effect on alopecia areata. Utilizing the Youden index to optimize specificity, we propose that IgG4 levels below 824.85 mg/L and IFN-γ levels above 0.565 pg/mL could serve as biomarkers for assessing the risk of alopecia areata.

**Conclusions:** These findings highlight the need for further exploration of the link among alopecia areata, IgG4- and IFN-γ-related mechanisms, potentially uncovering novel therapeutic targets for managing this condition.

## INTRODUCTION

Alopecia areata is a common hair follicle disorder, with environmental factors playing a significant role in its pathogenesis (*Minokawa, Sawada & Nakamura, 2022*; *Paus, 2020*). This immune-mediated condition is characterized by an autoimmune response targeting hair follicles, leading to non-scarring alopecia. It typically presents as circular bald patches on the scalp (*Strazzulla et al., 2018*). While some individuals may report mild itching or tingling sensations before hair loss, many patients remain asymptomatic. The onset of alopecia areata is not gender-biased, indicating that both men and women are equally susceptible to the condition (*Mirzoyev et al., 2014*). However, it has been influenced by religious and cultural practices with women or men being more likely to pursue medical intervention depending on their cultural context. Approximately 60% of individuals affected by alopecia areata are likely to experience the onset of the condition between the age of 10 and 25 years (*Juárez-Rendón et al., 2017*).

Hair follicles are believed to evade autoimmune responses by enhancing inhibitory signals in their microenvironment (*Paus, Bulfone-Paus & Bertolini, 2018*). This mechanism may disturb or suppress the activity of CD8+ cells, a specific type of immune cell, and natural killer (NK) cells (*Ito et al., 2008*). In contrast, patients with alopecia areata exhibit an infiltration of CD56+ NKG2D+ NK cells around the hair follicles. This indicates that the response of NK cells in alopecia areata patients may be enhanced or altered, differing from the normal immune response. Thus, numerous studies have also reported elevated levels of various cytokines in alopecia areata patients. The involvement of cytokines in alopecia areata is intricate. Transcriptomic analysis reveals that the alopecia areata signature, characterized by the upregulation of Th1, Th2, IL-9/Th9, and IL-23 cytokines, was prominently associated with the disease (*Suárez-Fariñas et al., 2015*). Furthermore, across various stages of alopecia areata, there is an infiltration of Th17 cells around the hair follicles (*Tanemura et al., 2013*). Similarly, the concentration of serum IL-17A, a proinflammatory cytokine characteristic of the Th17 subset, has been determined to be markedly higher in individuals suffering from alopecia areata (*El-Morsy et al., 2016*). In addition, others studies have documented an increase in eosinophils, mast cells, and serum IgE levels among patients with alopecia areata, highlighting the potential immunological underpinnings of the condition (*Ito et al., 2020*; *Yoon et al., 2014*; *Zhang et al., 2015*; *Zhao et al., 2012*). On the other hand, unhealthy lifestyle habits also contribute to alopecia areata. Smoking, drinking, and shift work can disrupt immune function, elevate inflammatory cytokines, and indirectly exacerbate the condition (*Ando et al., 2015*; *Dai et al., 2020*; *Sawada et al., 2021*). For instance, smoking and drinking are associated with increased production of IL-13, IFN-γ, TGF-α, and IL-6 (*Feleszko et al., 2006*; *Ockenfels et al., 1996*; *Sørensen et al., 2017*), while obesity induced by overeating may promote IL-17 production (*Nakamizo et al., 2017*), further elevating the risk of alopecia areata.

Current treatments for alopecia areata, like corticosteroids and immunomodulators, have limited efficacy and can lead to adverse reactions and high recurrence rates, especially in severe cases (*Zhou et al., 2021*). However, advances in understanding alopecia areata's pathogenesis have paved the way for novel therapies like JAK inhibitors, which hold promise for more effective and personalized interventions (*Chim et al., 2024*). Moreover, deficiency of vitamin D has been identified in alopecia areata patients (*Liu et al., 2020*). Therefore, providing appropriate amounts of vitamin D is considered a beneficial treatment for alopecia areata patients.

In this study, we analyzed the serum profiles of alopecia areata patients from Fujian, China. We examined cytokines, eosinophils, vitamin levels, total IgG, IgG4, and IgE in both patients and healthy individuals. By statistically evaluating the profiles among immune factors and vitamin levels, we aimed to identify specific serum indicators that can effectively differentiate alopecia areata patients, providing insights into its etiology in this region.

## MATERIALS AND METHODS

### Participant recruitment

The Ethics Committee of The Second Affiliated Hospital of Xiamen Medical College approves to carry out the study within its facilities (approval number: NO. 2022038). The informed consent was obtained from all participants. The consent process involved providing patients with detailed information about the study, including its purpose, procedures, potential risks and benefits, and their rights to withdraw at any time without penalty. Consent was documented in a written form, with patients signing a article-based consent form during their visit to the clinic. This approach ensured that all participants were fully aware of and agreed to the terms of the study prior to their involvement. This study on alopecia areata recruited patients from the dermatology outpatient clinic at the Second Affiliated Hospital of Xiamen Medical College, Fujian, China, between December 2023 and July 2024, amounting to a total of 28 cases. Only patients with patch-type alopecia areata were included in this study. Inclusion criteria required patients to meet the diagnostic standards for alopecia areata, defined by the sudden onset of smooth, hairless patches without trauma or other conditions that could cause hair loss (Fig. 1). Cases of pseudopelade, cicatricial alopecia, and other forms of hair loss were rigorously excluded. Participants were also required to abstain from systemic corticosteroids or immunosuppressants for at least 2 months before enrollment and to avoid topical corticosteroids for 1 month prior. Additionally, no medications aimed at promoting hair growth were used. Both the alopecia areata group and the healthy control group were screened to exclude individuals with a history of autoimmune, immunological, systemic, or psychiatric disorders.

### Blood analysis

The blood samples were entrusted to Xiamen KingMed Diagnostics Group Co., Ltd. (Fujian, China) for a comprehensive analysis of cytokines, eosinophils, water-soluble

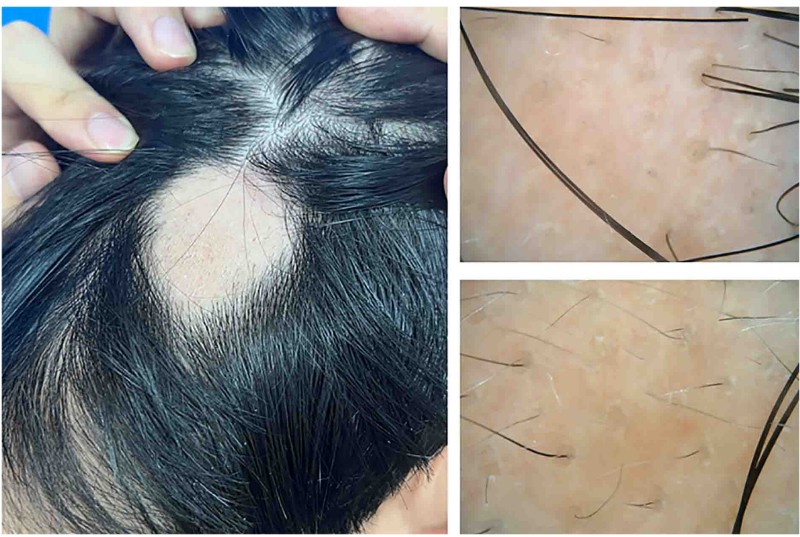

**Figure 1 Hair appearance and dermoscopic features of the patient with alopecia areata.**

vitamins, fat-soluble vitamins and immunoglobulins. Cytokines, including IL-2, IL-4, IL-6, IL-10, TNF-α, and IFN-γ, were quantified using the iMatrix Liquid Chip Platform (Joinstar Biomedical Technology Co., Ltd., Zhejiang, China) and the Multiple Cytokines (12-items) Detection Kit (Boshengte Biotechnology Co., Ltd., Chongqing, China). Eosinophils were analyzed using the Mindray BC-6800PLUS (Mindray Bio-Medical Electronics Co., Ltd., Shenzhen, China) with reagent kits including M-68PLH, M-68PLD, M-68PLN, M-68PFN, M-68PFD, and DS DILUENT kits (Mindray Bio-Medical Electronics, Shenzhen, China). Vitamins (water- and fat-soluble) were analyzed using an ultra-high-performance liquid chromatography system (Agilent 1290 II; Agilent Technologies, Inc., CA, USA) and a high-performance liquid chromatography-triple quadrupole mass spectrometer (API 5500 and API 4000) (AB SCIEX LLC, MA, USA). IgG was measured using the Mindray BS-2800M analyzer (Mindray Bio-Medical Electronics, Shenzhen, China) with the Immunoglobulin G Kit (Mindray Bio-Medical Electronics, Shenzhen, China). IgG4 was conducted using the optimised protein system with the Optilite IgG4 Kit (Binding Site Group Ltd., Birmingham, UK). IgE was analyzed using the Kaeser 6600 Automated Chemiluminescence Immunoassay Analyzer (Keysmile Biological Technology Co., Ltd., Chongqing, China) with the TIgE Diagnostic Kit (CLIA) (Kangrun Biotech Co., Ltd., Guangzhou, China).

## Statistical analysis

Wilcoxon rank sum test was used to determine statistical significance with Mann-Whitney U (M), performed using IBM SPSS Statistics (SPSS Inc., Chicago, IL, USA). Principal component analysis (PCA) was conducted *via* the Wei Sheng Xin online platform (https://www.bioinformatics.com.cn), a specialized resource for data analysis and visualization (*Tang et al., 2023*).

**Table 1  Basic information of alopecia areata patients and healthy individuals, including age, height, weight and blood pressure.** F and M indicate the numbers of female and male, respectively. Control indicates the healthy individuals.

| Treatments | Number of people | Age-years | Height (cm) | Weight (kg) | Blood pressure (mmHg) | |
|---|---|---|---|---|---|---|
| | | | | | Systolic pressure | Diastolic pressure |
| Alopecia areata | 28 (F: 11, M: 17) | 11~69 (29.1[a] ± 12.3[b]) | 147~184 (166.6 ± 8.8) | 35~174 (71.8 ± 28.2) | 100~136 (120.6 ± 10.1) | 65~91 (78.0 ± 6.0) |
| Control | 28 (F: 11, M: 17) | 20~48 (29.6 ± 6.8) | 150~184 (169.0 ± 9.6) | 40~96 (66.0 ± 14.4) | 93~144 (118.3 ± 10.3) | 60~92 (75.0 ± 8.8) |

**Notes:**
[a] The data presents the mean values of age, height, weight, and blood pressure.
[b] The data presents the standard deviations of age, height, weight, and blood pressure.

## RESULTS

### Participants and their blood analysis

A comparative analysis was conducted between patients diagnosed with alopecia areata and a control group of healthy individuals. Both groups included 28 participants, with equal representation of females (11) and males (17) in each group (Table 1). The age range for the alopecia areata group was 11–69 years, with a mean age of 29.1 years. Heights varied from 147 to 184 cm (mean: 166.6 cm), and weights ranged from 35 to 174 kg (mean: 71.8 kg). For the healthy individuals, ages ranged from 20–48 years, with a mean age of 29.6 years. Heights spanned 150–184 cm (mean: 169.0 cm), and weights ranged from 40–96 kg (mean: 66.0 kg).

The blood analysis for cytokines, eosinophils, water-soluble vitamins, fat-soluble vitamins, and immunoglobulins, was conducted. The cytokine levels revealed notable distinctions between the alopecia areata group and the control group, though some parameters displayed overlapping tendencies. In the alopecia areata group, IL-2 levels averaged 0.365 ± 0.146 pg/mL, with a median of 0.315 pg/mL, closely resembling the control group, which exhibited an average of 0.367 ± 0.312 pg/mL and a median of 0.260 pg/mL (Fig. 2). IL-4 levels were noticeably elevated in the alopecia areata group, with an average of 2.195 ± 4.366 pg/mL and a median of 0.880 pg/mL, compared to 1.105 ± 0.766 pg/mL and a median of 0.790 pg/mL in the control group. Conversely, IL-6 levels were higher on average in the control group at 8.315 ± 8.367 pg/mL, with a median of 3.910 pg/mL, compared to the alopecia areata group, where the average was 3.672 ± 1.016 pg/mL with a median of 3.315 pg/mL. IL-10 levels varied widely in both groups; however, the alopecia areata group indicated higher average levels (6.150 ± 16.510 pg/mL) compared to the control group (4.015 ± 6.267 pg/mL), though median values were similar at 1.525 pg/mL and 1.655 pg/mL, respectively. Further differences were evident in TNF-α and IFN-γ levels. The alopecia areata group displayed slightly higher TNF-α levels, with an average of 0.775 ± 0.343 pg/mL and a median of 0.735 pg/mL, compared to the control group, where the average was 0.679 ± 0.325 pg/mL and the median 0.575 pg/mL. IFN-γ levels showed low variation, with averages of 0.591 ± 0.181 pg/mL in the alopecia areata group and 0.553 ± 0.351 pg/mL in the control group, while medians were higher in the alopecia areata group at 0.635 pg/mL *vs.* 0.520 pg/mL in the controls. PCA highlighted the

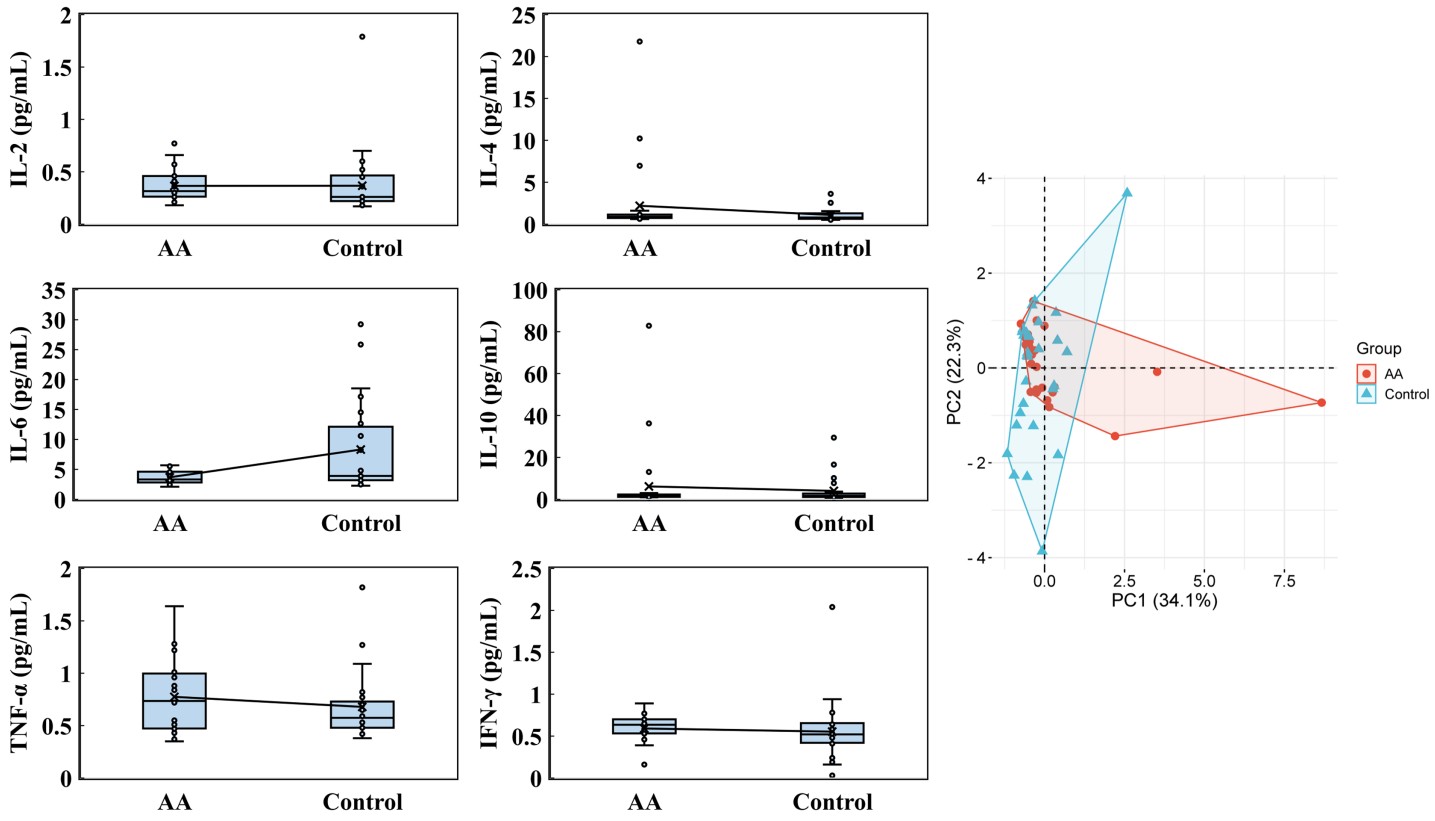

**Figure 2** **Cytokines analysis.** IL-2, IL-4, IL-6, IL-10, TNF-α, and IFN-γ were conducted by using the iMatrix 100 flow cytometry luminescence. PCA was analyzed by the online platform Wei Sheng Xin (https://www.bioinformatics.com.cn). AA and control indicate alopecia areata patients and healthy individuals, respectively.

cytokine profile differences between the groups, with PC1 accounting for 34.1% and PC2 contributing 22.3% of the variability, underscoring the importance of these cytokines in distinguishing the immune profiles of the two groups.

Regarding the eosinophils analysis, only slight variations were observed (Fig. 3). In the alopecia areata group, the average eosinophil count was $0.152 \pm 0.108$ ($10^9$ cells/L), with a median of 0.120, compared to an average of $0.172 \pm 0.142$ and a median of 0.125 in the control group. Similarly, the percentage of eosinophils (EO%) was nearly identical between the groups, with an average of $2.746 \pm 1.901$ and a median of 2.100 in the alopecia areata group, and $2.782 \pm 2.166$ with a median of 2.150 in the control group. Principal Component Analysis (PCA) revealed that the first principal component (PC1) accounted for 97.5% of the data variability in eosinophil levels, demonstrating its potential as a distinguishing metric. However, these findings suggest that eosinophil levels alone may not serve as a strong differentiating factor between the two groups.

Water-soluble vitamin analysis included vitamin B2, B3, B5, B6, B7, B12, folate acid and 5-MTHF (5-Methyltetrahydrofolate). Notably, vitamin B7, B12, and folate acid levels were largely undetectable in both groups, falling below the minimum detectable levels of 0.08 ng/mL, 0.10 ng/mL, and 0.44 ng/mL, respectively. PCA showed that PC1 and PC2

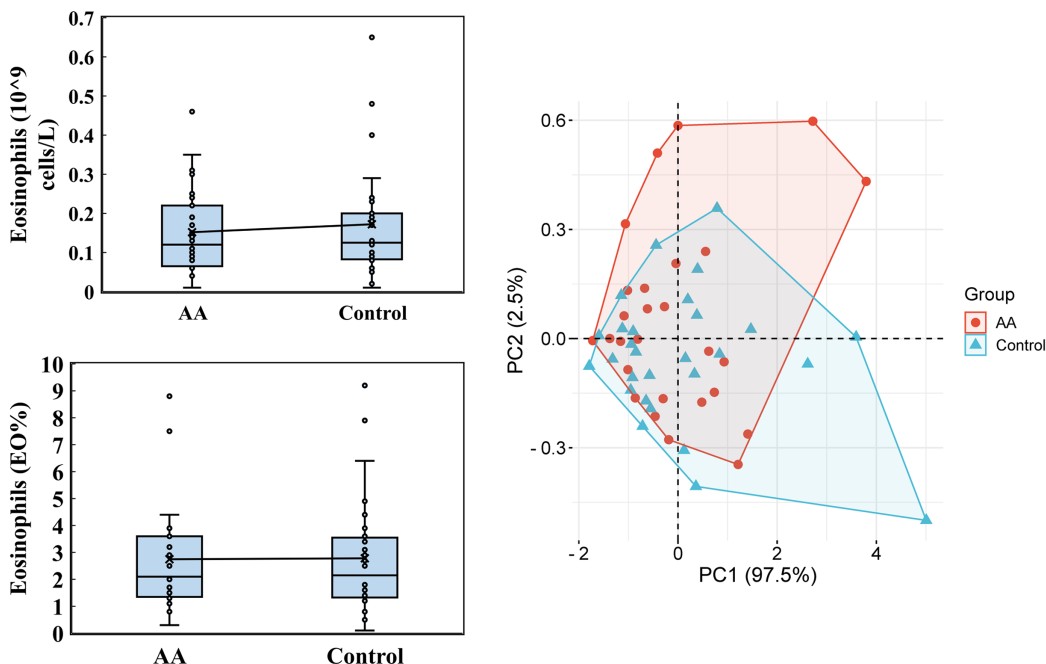

**Figure 3 Eosinophils analysis.** Eosinophils, encompassing both cell count and percentage, were performed by using the Mindray BC-6800PLUS automated hematology analyzer. PCA was analyzed by the online platform Wei Sheng Xin (https://www.bioinformatics.com.cn). AA and control indicate alopecia areata patients and healthy individuals, respectively.

accounted for 33.1% and 20.7% of the variability, respectively, underscoring the contribution of these vitamins to the observed differences (Fig. 4). For vitamin B2, the alopecia areata group had an average level of 12.926 ± 10.722 ng/mL with a median of 8.990, slightly lower than the control group's average of 13.241 ± 13.636 ng/mL and a median of 8.415. Vitamin B3 exhibited a contrasting pattern, where the alopecia areata group had a higher average of 68.0 ± 40.777 ng/mL and a median of 48.945 compared to the control group's average of 58.281 ± 19.636 ng/mL but a higher median of 57.320. Additionally, vitamin B5 levels were marginally lower in the alopecia areata group, with an average of 43.743 ± 9.776 ng/mL and a median of 41.630, compared to the control group's average of 47.894 ± 15.110 ng/mL and a median of 47.825. Vitamin B6 levels were generally low in both groups, with the alopecia areata group showing an average of 4.854 ± 9.741 ng/mL and a median of 2.185, slightly higher than the control group's average of 4.199 ± 8.072 ng/mL and median of 2.515. Lastly, for 5-MTHF, the alopecia areata group exhibited an average level of 7.603 ± 7.749 ng/mL and a median of 5.405, while the control group had a lower average of 7.286 ± 5.226 ng/mL but a higher median of 6.420, reflecting the subtle complexities in the distribution of these water-soluble vitamins across the two groups.

The analysis of fat-soluble vitamins, including vitamin A, E, K1, 25-hydroxyvitamin D2 (25-OH-VD2), 25-hydroxyvitamin D3 (25-OH-VD3), and 25-OH-VD (D2+D3), revealed that 25-OH-VD2 levels were consistently below the detection threshold of 2.2 ng/mL in

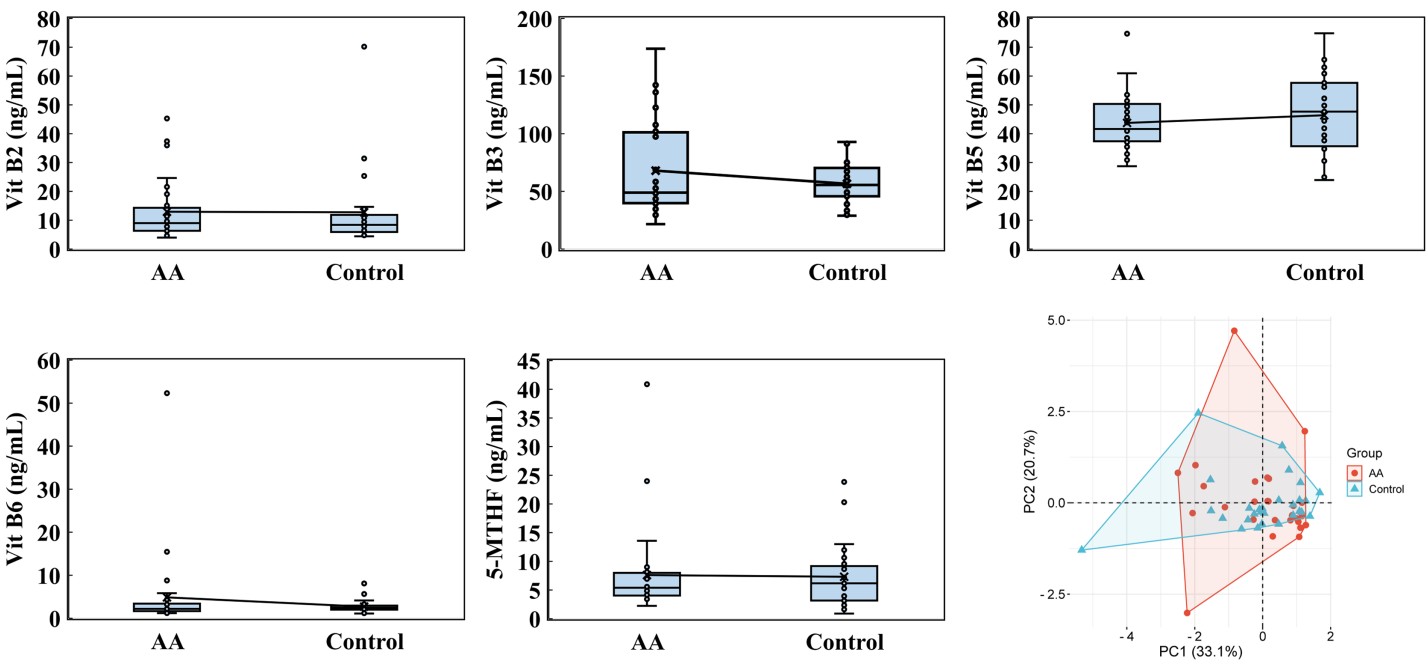

**Figure 4 Water-soluble vitamin analysis.** Vitamin B2, vitamin B3, vitamin B5, and 5-MTHF were performed by using a high-performance liquid chromatography with a triple quadrupole mass spectrometer. PCA was analyzed by the online platform Wei Sheng Xin (https://www.bioinformatics.com.cn). AA and control indicate alopecia areata patients and healthy individuals, respectively.

both the alopecia areata and control groups. PCA analysis highlighted the significant contribution of PC1 (47.5%) and PC2 (26.9%) to the variability in vitamin profiles, emphasizing differences in vitamin A, E, K1, 25-OH-VD3, and 25-OH-VD (D2+D3) levels (Fig. 5). For vitamin A, the alopecia areata group demonstrated an average concentration of 0.540 ± 0.127 mg/L with a median of 0.550, slightly higher than the control group's average of 0.501 ± 0.144 mg/L and median of 0.480. Vitamin E levels were slightly lower in the alopecia areata group, averaging 10.539 ± 2.138 mg/L with a median of 10.900, compared to the control group's higher average of 10.782 ± 2.621 mg/L and median of 10.300. Vitamin K1 levels showed similar result, with the alopecia areata group recording an average of 1.145 ± 0.668 ng/mL and a median of 1.000, while the control group had an average of 1.144 ± 0.865 ng/mL and median of 0.790. The concentrations of 25-OH-VD3 were found to be quite similar between the alopecia areata group and the control group. The alopecia areata group had an average of 18.543 ± 6.011 ng/mL and a median of 17.050 ng/mL, which was not significantly different from the control group's average of 18.168 ± 6.106 ng/mL and median of 16.400 ng/mL. Similarly, the levels of 25-OH-VD (D2+D3) showed minimal variation between the two groups, with averages and medians that were almost indistinguishable, reflecting a consistency in these vitamin levels across the cohorts.

Notable differences emerged in immunoglobulin levels, particularly IgG4. While the average IgE level in the alopecia areata group was 116.084 ± 152.834 IU/mL, with a median of 52.715 IU/mL, the control group showed a higher average of 130.331 ± 213.136 IU/mL

PeerJ

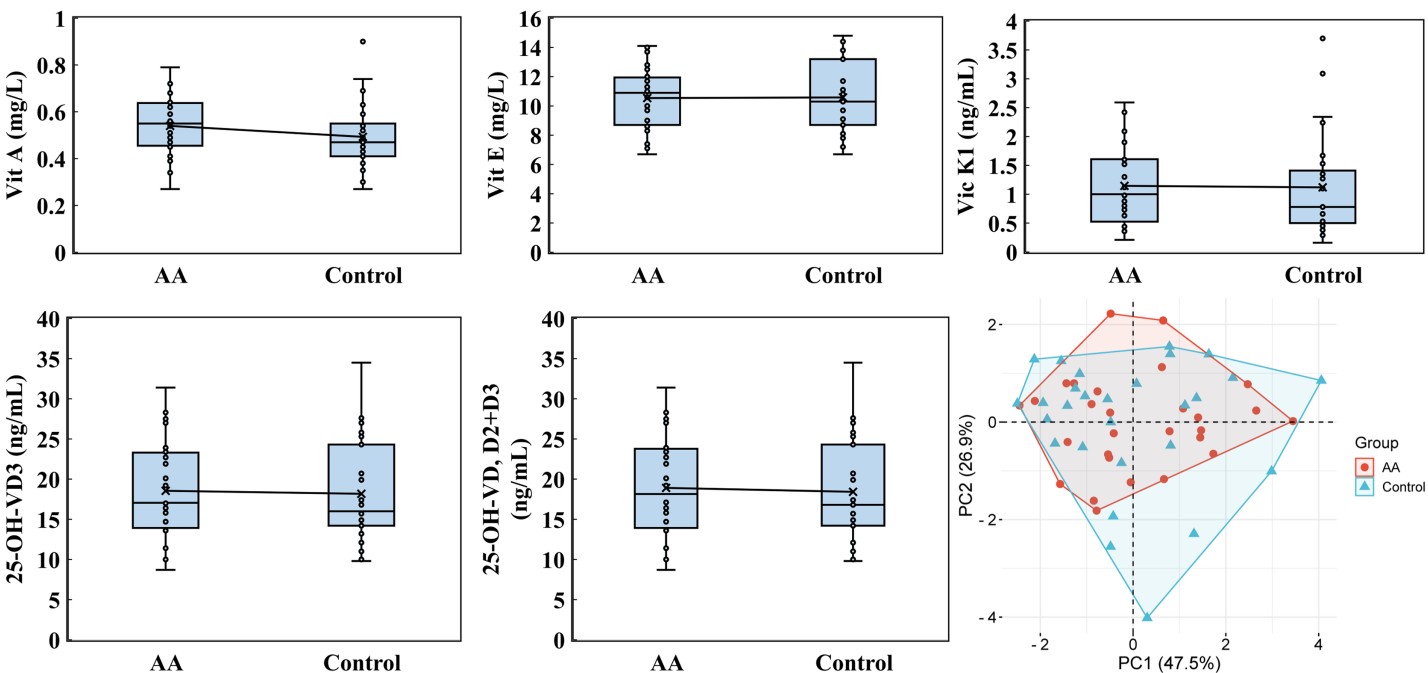

**Figure 5** **Fat-soluble vitamin analysis.** Vitamin A, vitamin E, vitamin K1, 25-OH-VD3, and 25-OH-VD were performed by utilizing an ultra-high-performance liquid chromatography and a high-performance liquid chromatography with a triple quadrupole mass spectrometer. PCA was analyzed by the online platform Wei Sheng Xin (https://www.bioinformatics.com.cn). AA and control indicate alopecia areata patients and healthy individuals, respectively.                                                                            

but a lower median of 44.505 IU/mL (Fig. 6). For IgG, the alopecia areata group presented an average level of 11.834 ± 2.343 g/L with a median of 11.960, compared to 13.130 ± 2.457 g/L and a median of 13.125 in the control group. Strikingly, IgG4 levels in the alopecia areata group were significantly decreased, averaging 604.071 ± 627.090 mg/L with a median of 515.950 mg/L, whereas the control group exhibited much higher levels, with an average of 898.186 ± 643.939 mg/L and a median of just 845.000 mg/L. PCA further underscored the importance of these findings, with PC1 explaining 47.8% of the variability in immunoglobulin data, highlighting its critical role in distinguishing the two groups. These differences, particularly in IgG4, suggest a potential immunological marker for alopecia areata, warranting further investigation.

## Wilcoxon rank sum test

Wilcoxon rank sum test was conducted to predict disease risk, support clinical decision-making, and analyze the association of various biomarkers with alopecia areata. This approach provided insights into the factors influencing health outcomes (*Lin et al., 2023*). Therefore, it was used to observe the significance analysis of cytokines, eosinophils, water-soluble vitamins, fat-soluble vitamins, and immunoglobulins in alopecia areata. The findings revealed significantly different levels of IgG4 and IFN-γ between alopecia areata patients and healthy individuals (Table 2). In individuals with alopecia areata, IgG4 levels tended to be lower compared to those in healthy individuals, whereas IFN-γ levels

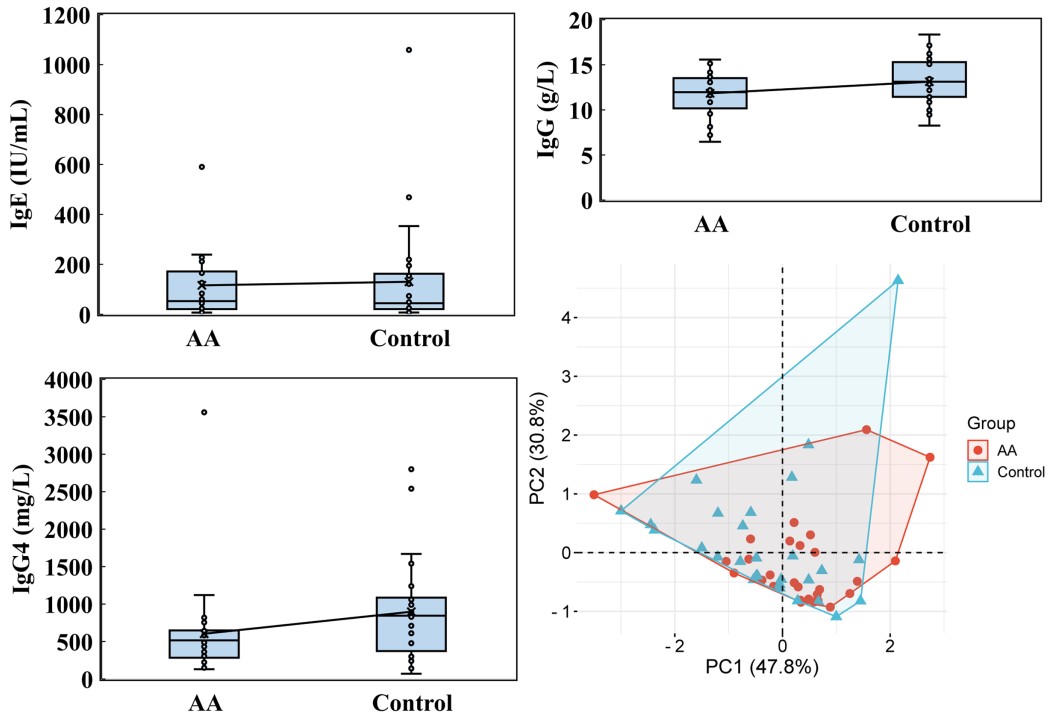

**Figure 6 Immunoglobulin analysis.** IgE, IgG, and IgG4 levels were performed by utilizing Kaeser 6600 Automated Chemiluminescence Immunoassay Analyzer, Mindray BS-2800M analyzer, and optimised protein system. PCA was analyzed by the online platform Wei Sheng Xin (https://www.bioinformatics.com.cn). AA and control indicate alopecia areata patients and healthy individuals, respectively.

**Table 2 Significance analysis of immunoglobulin, water-soluble vitamins, fat-soluble vitamins and cytokines in the blood of patients with alopecia areata and healthy individuals using Wilcoxon rank sum test.**

| Blood assay | Variations | Mann-Whitney U[a] | Z[b] | p-value |
|---|---|---|---|---|
| Eosinophils | Eosinophils ($10^9$ cells/L) | 369.500 | −0.369 | 0.712 |
| | Eosinophils (EO%) | 375.500 | −0.271 | 0.787 |
| Immunoglobulins | IgE | 384.000 | −0.131 | 0.896 |
| | IgG4 | 232.000 | −2.622 | 0.009 |
| | IgG | 284.500 | −1.762 | 0.078 |
| Water-soluble vitamins | Vitamin B2 | 375.000 | −0.279 | 0.781 |
| | Vitamin B3 | 382.000 | −0.164 | 0.870 |
| | Vitamin B5 | 325.500 | −1.090 | 0.276 |
| | Vitamin B6 | 363.000 | −0.475 | 0.635 |
| | 5-MTHF | 367.000 | −0.410 | 0.682 |
| Fat-soluble vitamins | Vitamin A | 308.000 | −1.377 | 0.168 |
| | Vitamin E | 385.500 | −0.107 | 0.915 |
| | 25-OH-VD3 | 380.500 | −0.188 | 0.851 |
| | 25-OH-VD, D2+D3 | 381.500 | −0.172 | 0.863 |
| | Vitamin K1 | 366.000 | −0.426 | 0.670 |

| Table 2 (continued) | | | | |
|---|---|---|---|---|
| Blood assay | Variations | Mann-Whitney U[a] | Z[b] | p-value |
| Cytokines | IL-2 | 297.000 | −1.558 | 0.119 |
| | IL-4 | 331.000 | −1.000 | 0.317 |
| | IL-6 | 277.500 | −1.877 | 0.061 |
| | IL-10 | 391.000 | −0.016 | 0.987 |
| | TNF-α | 324.000 | −1.115 | 0.265 |
| | IFN-γ | 269.500 | −2.008 | 0.045 |

Notes:
[a] Mann-Whitney U: A lower U value indicates a more significant difference between the two samples.
[b] Z: A higher Z-value (whether positive or negative) indicates a more significant difference between the two samples.

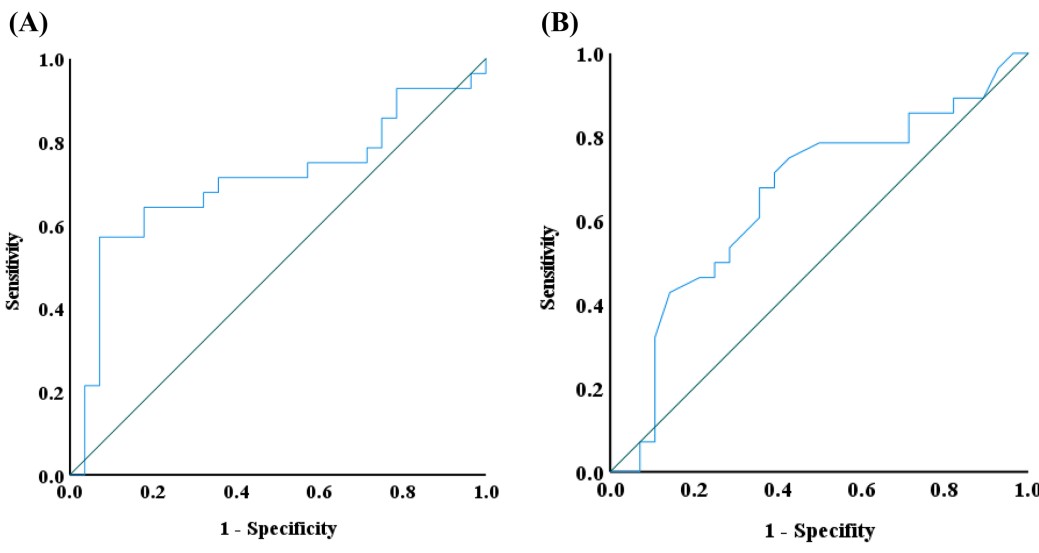

**Figure 7 Receiver operating characteristic (ROC) curve analysis.** ROS analysis of (A) IgG4 and (B) IFN-γ was conducted by using IBM SPSS statistics.

exhibited an opposite, increasing trend. Consistent with the pattern observed in IgG4, patients with alopecia areata also exhibited a tendency towards lower IgG levels, although this association only reached marginal statistical significance.

## Receiver operating characteristic (ROC) curve analysis

The ROC curve is a diagnostic tool used to evaluate the performance of binary classification methods (*Nahm, 2022*). It illustrates the trade-off between the true positive rate (sensitivity) and the false positive rate (1-specificity) across various thresholds. Given the Wilcoxon rank sum test results, IgG4 and IFN-γ emerged as the significant biomarkers. Therefore, a ROC curve was applied to assess its diagnostic utility. The area under the ROC curve (AUC) was calculated to measure the discrimination ability of IgG4 and IFN-γ between the two groups according to the SPSS Statistics analysis. An AUC closer to 1 indicates strong discrimination, while an AUC below 0.5 suggests no discrimination, equivalent to random guessing. The AUC for IgG4 and IFN-γ was 0.704 and 0.656,

respectively, indicating a moderate discriminatory effect in distinguishing alopecia areata patients from controls (Fig. 7). Additionally, the Youden index identified an IgG4 threshold of 824.85 mg/L as a potential indicator of alopecia areata risk. Nevertheless, the Youden index of IFN-γ was in the range of 0.535–0.565 pg/mL.

## DISCUSSION

This study aimed to investigate differences in immune factors and vitamin concentrations between individuals with alopecia areata and a healthy control from Fujian Province, China, to better understand how these variations may influence the etiology of alopecia areata. By conducting a comprehensive statistical analysis of blood samples, we identified key findings that could contribute to a deeper understanding of the pathophysiological mechanisms underlying this condition.

Previous studies have established that levels of specific cytokines in alopecia areata patients are often elevated compared to healthy individuals. Notably, cytokines such as IL-6, IL-13, IL-23, IL-17A, IFN-γ, and TGF-α have been highlighted in prior research (*El-Morsy et al., 2016*; *Feleszko et al., 2006*; *Ockenfels et al., 1996*; *Sørensen et al., 2017*; *Suárez-Fariñas et al., 2015*). Expanding on this, our study examined a broader range of cytokines, including IL-2, IL-4, IL-6, IL-10, IFN-γ, and TNF-α, to evaluate their potential correlation with the onset and progression of alopecia areata in blood assays (Fig. 2). Previous studies have indicated that patients with alopecia areata exhibit increased IL-2 mRNA expression in the deep dermis surrounding hair follicles, with particularly high levels in active lesional areas (*Lensing & Jabbari, 2022*). Experimental mouse models further underscore the pivotal role of IL-2 and IFN-γ in the pathogenesis of the disease (*Freyschmidt-Paul et al., 2005*, *2006*). Additionally, elevated levels of IL-4, IL-6, and IL-10 in alopecia areata patients have been reported (*Aşkın et al., 2021*; *Gautam et al., 2020*; *Lensing & Jabbari, 2022*; *Muhammad & Pirzado, 2024*; *Šutić Udović et al., 2024*). These cytokines are thought to play critical roles in the disease's initiation and progression, and often interpreted as markers of immune system dysregulation. In this study, IFN-γ levels were significantly elevated in patients with alopecia areata, with a *p*-value less than 0.05, indicating a statistically significant difference (Table 2). Based on the analysis of the ROC curve, the concentration of IFN-γ may serve as a predictive marker for the risk of alopecia areata when it exceeded 0.565 pg/mL (Fig. 7). Drawing from previous research, IFN-γ is not only indicative of alopecia areata but also plays a pivotal role in the pathogenesis of alopecia universalis, which is considered an advanced form of alopecia areata (*Perricone et al., 2024*). Therefore, how to regulate the expression of IFN-γ has become a target option for treating alopecia areata, such as using JAK inhibitors. Such findings also point to a complex interaction between immune system dysfunction and disease development, underscoring the need for further research into the role of cytokines in autoimmune responses.

On the other hand, IL-4 and IL-13 are essential signaling molecules that activate and promote the proliferation of eosinophils (*Nakagome & Nagata, 2024*). These cytokines, released rapidly upon exposure to allergens or pathogens, bind to eosinophil receptors to trigger their effector functions. These functions include the synthesis of eosinophil granule

proteins, the release of chemical mediators, and active participation in inflammatory responses. Numerous studies involving alopecia areata patients have consistently reported elevated eosinophil levels (*Yoon et al., 2014*). However, in the alopecia areata patients we diagnosed, we did not observe a higher trend of eosinophils (Fig. 3). This discrepancy may be due to our analysis focusing on peripheral blood eosinophils rather than those localized near hair follicles, which made the difference in peripheral blood less obvious.

Our research further explored the potential link between vitamin deficiencies and alopecia areata. Previous studies have suggested a strong correlation between vitamin D deficiency and alopecia areata (*Liu et al., 2020*). Additionally, animal studies suggest that vitamin A levels may modulate the onset of alopecia areata, with both excessive and insufficient levels potentially exacerbating the condition (*Thompson et al., 2017*). Building on this, we conducted a comprehensive analysis of both water-soluble and fat-soluble vitamins (Figs. 4 and 5). In this study, while vitamin A levels in alopecia areata patients tended to be higher on average, the difference did not reach statistical significance ($p = 0.168$). Nevertheless, this observed trend warrants further investigation to clarify its potential role in disease pathogenesis. Intriguingly, our findings revealed deficiencies in vitamin B7, B12, folic acid, and 25-OH-VD2, not only in alopecia areata patients but also in healthy controls. Deficiencies in these vitamins are known to elevate the risk of conditions such as oral candidiasis, chronic kidney disease, increased inflammation, oxidative stress, osteoporosis, and immune system dysfunction (*Lešić et al., 2024*; *Rakovac & Sajković, 2023*; *Wu, McDonnell & Chinnadurai, 2023*). These results highlight the need for an in-depth examination of dietary and lifestyle habits within this population, providing local health authorities with critical data for devising targeted public health policies and intervention strategies.

In analyzing cytokine and vitamin levels, we observed that individuals with alopecia areata exhibited elevated average levels of IL-4, IL-10, TNF-α, and IFN-γ, surpassing those of healthy controls (Fig. 2). Conversely, the alopecia areata group displayed reduced levels of vitamin B2, vitamin B5, and vitamin E compared to the control group (Figs. 4 and 5). These findings align with previous studies on alopecia areata. However, when examining the distribution and variability of cytokine and vitamin levels, they alone proved insufficient to differentiate alopecia areata patients from healthy individuals, with the notable exception of IFN-γ. In contrast, a statistically significant difference was observed in the expression of immunoglobulin IgG4 (Table 2). Patients with alopecia areata demonstrated significantly lower IgG4 levels than the control group, with a *p*-value of 0.009. Similarly, a decreasing trend in total IgG levels had been noted among patients with alopecia areata, although this trend did not reach conventional statistical significance (*p*-value 0.078). Nonetheless, the observed reduction in these antibodies merits our attention and further investigation. IgG4 is a distinct subclass of immunoglobulin G, comprising only 3–6% of the total IgG pool. This specificity makes IgG4 a unique component within the IgG family. IgG4-related disease (IgG4-RD) is a systemic fibroinflammatory condition affecting various organ systems, including the bile ducts, lacrimal glands, lymphatic vessels, major salivary glands, pancreas, and retroperitoneal space (*Sánchez-Oro, Alonso-Muñoz & Martí Romero, 2019*). In the pulmonary system,

IgG4-RD may present as inflammatory pseudotumor, organizing pneumonia, interstitial pneumonitis, or lymphomatoid granulomatosis (*Morales et al., 2019*). Consequently, IgG4-RD is regarded as a systemic illness that encompasses a spectrum of single-organ disorders. Shared characteristics of IgG4-RD include increased serum IgG4 concentrations, neoplastic-like organ swelling, distinctive histopathology, and specific immunostaining features (*Matsui, 2019*; *Sánchez-Oro, Alonso-Muñoz & Martí Romero, 2019*). Moreover, a case of autoimmune pancreatitis (an IgG4-related condition) was accompanied by symptoms of alopecia, with perifollicular infiltration of IgG4-positive cells (*Ikeda, Kaminaka & Furukawa, 2019*). Another case involving generalized alopecia and severe atopic dermatitis revealed fibrosing inflammation between hair follicles and IgG4-positive plasma cells in a scalp biopsy (*Kossard, Sheriff & Murrell, 2021*). These cases hint at a potential link between elevated IgG4 levels and alopecia areata. However, the deficiency or reduction in IgG4 levels can also be implicated in the development of severe conditions, including chronic obstructive pulmonary disease (COPD), inflammatory bowel disease (IBD), follicular bronchiolitis, and cartilage-hair hypoplasia (CHH) (*Assaad, Aqeel & Walsh, 2022*; *Koutroumpakis et al., 2021*; *Lee et al., 2022*; *Mäkitie, Kaitila & Savilahti, 2000*). CHH, a form of metaphyseal chondrodysplasia, is typically characterized by compromised cellular immunity (*Mäkitie, Kaitila & Savilahti, 2000*). Among its features, the hair hypoplasia or alopecia is a notable characteristic of this disease. Patients with cartilage-hair hypoplasia who have symptoms of high incidence of Hirschsprung disease also experience symptoms of alopecia and alopecia areata (*Mäkitie, Kaitila & Rintala, 2001*). While the mechanism between IgG4 and alopecia areata is rarely documented, emerging evidence suggests that IgG4-related diseases or IgG4 deficiency can affect not only internal organs but also the skin. In our study, although the IgG4 levels in alopecia patients fell within the normal parameters, there was a significant difference in the decreasing trend compared with healthy individuals. ROC curve analysis was employed to decline IgG4 as a diagnostic marker for alopecia areata. The ROC curve indicated that an IgG4 threshold of 824.85 mg/L could serve as a diagnostic indicator (Fig. 7). The AUC of 0.704 suggests moderate diagnostic efficacy. This analysis is crucial for determining optimal cut-off values, benchmarking the performance of diagnostic markers, and refining diagnostic tools.

## CONCLUSIONS

Although the variations in cytokines and vitamins such as IL-2, IL-4, IL-6, IL-10, TNF-α, and vitamins in the blood are insufficient to differentiate alopecia areata, we have observed a notable increase in IFN-γ and decrease in IgG4 levels among patients with this condition. Utilizing the analysis of the ROC curve, we can propose the IFN-γ and IgG4 concentration index that may serve as a diagnostic tool for identifying alopecia areata. The elevation of IFN-γ observed in our study aligned with previous research, indicating a consistent presence of these cytokines among patients suffering from alopecia areata. Interestingly, both IgG and IgG4 concentrations exhibited a lower trend in the blood of alopecia areata patients, with IgG4 demonstrating a particularly significant difference. The finding emphasizes the need for further research into the role of IgG4 in the pathogenesis of

alopecia areata. Future studies should focus on elucidating the precise mechanisms by which IgG4 contributes to disease progression and exploring potential therapeutic targets to improve outcomes for individuals affected by alopecia areata. Additionally, developing multi-parameter diagnostic models by integrating these biomarkers with machine learning algorithms represents a promising research direction.

## ACKNOWLEDGEMENTS

We thank Mingjie Chen (Shanghai NewCore Biotechnology Co., Ltd) for providing data analysis and visualization support.

### Funding

This work was supported by the Engineering Research Center of Natural Cosmeceuticals College of Fujian Province Fund (Xiamen Medical College)(No. XMMC-NC202202). The funders had no role in study design, data collection and analysis, decision to publish, or preparation of the manuscript.

### Grant Disclosures

The following grant information was disclosed by the authors:
Engineering Research Center of Natural Cosmeceuticals College of Fujian Province Fund (Xiamen Medical College): XMMC-NC202202.

### Competing Interests

The authors declare that they have no competing interests.

### Author Contributions

- Jincheng Ke conceived and designed the experiments, performed the experiments, authored or reviewed drafts of the article, and approved the final draft.
- Fangfang Chen analyzed the data, prepared figures and/or tables, authored or reviewed drafts of the article, and approved the final draft.
- Yu-Pei Chen conceived and designed the experiments, analyzed the data, prepared figures and/or tables, authored or reviewed drafts of the article, and approved the final draft.
- Mingli Zhang performed the experiments, authored or reviewed drafts of the article, and approved the final draft.
- Li Ma performed the experiments, authored or reviewed drafts of the article, and approved the final draft.

### Human Ethics

The following information was supplied relating to ethical approvals (*i.e.*, approving body and any reference numbers):

Approval was granted by the Ethics Committee of The Second Affiliated Hospital of Xiamen Medical College (approval number: NO. 2022038).

## Data Availability

The raw data is available in the Supplemental Files.

## Supplemental Information

Supplemental information for this article can be found online at http://dx.doi.org/10.7717/peerj.19430#supplemental-information.

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
