# Peer review of "Unveiling the immune and vitamin profiles of blood: the potential biomarkers for alopecia areata"

_PeerJ, doi:10.7717/peerj.19430_

## Round 0.1 · original submission · Major Revisions

Dr Chen - as you will see from the reviewers' comments, there is need to review the manuscript to address as many of these as possible, most particularly the issue of data handling and statistical analysis.

Reviewer 1 ·

Basic reporting

no comment

Experimental design

no comment

Validity of the findings

no comment

Additional comments

The study included a small sample size and lacked innovation, and the results could not support the research conclusions

·

Basic reporting

The authors investigated the possibility of using immunological markers and vitamins in alopecia areata.

Experimental design

No

Validity of the findings

You can add more in the discussion for the role of vitamins in AA

Reviewer 3 ·

Basic reporting

The quality of academic writing in English is high. The manuscript is well-organized and easy to understand.

There are several places where reference is missing:
- Line 67-68: Hair follicles are believed to evade autoimmune responses by enhancing inhibitory signals in their microenvironment.
- Line 69-70: In contrast, patients with alopecia areata exhibit an infiltration of CD56+ NKG2D+ NK cells around the hair follicles.
- Line 87-88: obesity induced by overeating may promote IL-17 production, further elevating the risk of alopecia areata.
- Line 289-290: Experimental mouse models further underscore the pivotal role of IL-2 and IFN-3 in the pathogenesis of the disease.

The Table and Figure legend can be more specific, for example:
- Table 1: Please indicate what the numbers in the brackets refer to. For example, Age of AA is 29.1 +/- 12.3, is 29.1 mean or median? is 12.3 standard deviation? I know the authors have stated in their main text saying the mean age is 29.1, but reminding the authors again in the table legend will help the readers understand better.
- Figures 7: Please indicate in each heatmap, what each row represents. For example, in the last heatmap "immunoglobulin", each row should represent the concentration of IgE, IgG and IgG4. Please label them.

Experimental design

The authors have analyzed the concentration of at least 21 components and compared their level between control and patients. The authors identified differential components by performing Wilcoxon rank sum test for each component. In this case, the authors have conducted multiple comparisons which is prone to false positives in statistical testing. In addition to what they have done, the authors should perform statistical correction for multiple hypothesis testing, such as Bonferroni method or calculating a false-discovery rate.

Line 127-128, the authors wrote that they used a Flow Cytometry Luminescence Analysis Instrument to analyze cytokine level in the peripheral blood. I understand that the vendor probably has named their instrument with "flow cytometry" in a foreign language and thus it is translated into "flow cytometry" in English. I'm a little concerned here, as "cytometry" typically refers to analysis that is related to cells, like cell size, shape or level of protein that is *on or inside the cell*. Because the cytokines analyzed here seem to all be circulating in the blood and don't involve a cell component, I recommend the author to report the instrument name in a better way so that "cytometry" is not mentioned here.

Line 254: "IgG4 and IFN-gamma exhibited a statistically significant correlation with AA" - I recommend avoid using the word "correlation" here as it leads the readers to look for a correlation analysis. The authors may say they are significantly different between AA patients and control.

Validity of the findings

Because the statistical analysis performed here may be subject to multiple comparison error, the authors will need to incorporate a step to correct for this error, and then re-evaluate whether IgG4 and IFN-gamma are still statistically significant after that.

The authors have provided the raw data and I appreciate that.

·

Basic reporting

In the present study, the authors aimed to identify potential biomarkers for alopecia areata (AA) through blood analysis. This is an interesting study to explore the immune and vitamin profiles of blood in AA patients. However, several limitations should be addressed:

1. Detailed data for all variables (e.g., mean, standard deviation [SD], median, and interquartile range) should be added to Table 2 to enhance transparency.
2. AA has three clinical subtypes: patch-type alopecia areata (AAP), alopecia totalis (AT), and alopecia universalis (AU). The authors should analyze the differences in these biomarkers across AA subtypes to determine subtype-specific variations.
3.The rationale for using the Wilcoxon rank-sum test to determine statistical significance requires clarification. It is unclear whether the authors assessed the normality of the data distribution. If normality was not verified, the statistical methods should be re-evaluated.
4.The AUC values of IFN-γ (0.656) and IgG4 (0.704) for assessing AA risk suggest limited diagnostic utility. To improve accuracy, the authors could consider integrating these markers with machine learning algorithms to develop a multi-parameter diagnostic model.
5.Figure 7 appears redundant in the current context. Consider removing this figure to improve conciseness, or clarify its relevance to the main findings.

Experimental design

no comment

Validity of the findings

no comment

Additional comments

no comment

---

## Round 0.2 · accepted · Accept

Thank you for addressing the remaining reviewers' comments/suggestions satisfactorily.

Reviewer 3 ·

Basic reporting

No comment

Experimental design

No comment

Validity of the findings

No comment

Additional comments

The authors have successfully addressed my comments. The manuscript is suitable for publication.